# Real-World Budget Impact of Fidaxomicin versus Vancomycin or Metronidazole for In-Hospital Treatment of *Clostridioides difficile* Infection

**DOI:** 10.3390/antibiotics12010106

**Published:** 2023-01-06

**Authors:** Laura Whitney, John Nesnas, Timothy Planche

**Affiliations:** 1St George’s University Hospitals NHS Foundation Trust, London SW17 0QT, UK; 2Infection Care Group, St George’s University Hospitals NHS Foundation Trust, London SW17 0QT, UK; 3Institute of Infection and Immunity, St George’s University of London, London SW17 0RE, UK

**Keywords:** *Clostridioides difficile*, fidaxomicin, healthcare resource utilization, real-world outcomes, recurrence

## Abstract

Fidaxomicin, a macrocyclic antibiotic, selectively kills *Clostridioides difficile* and reduces *C. difficile* infection (CDI) recurrence compared with vancomycin, but some studies and guidelines report fidaxomicin as being less cost-effective. The aim of this study was to compare the cost-effectiveness and budget impact of fidaxomicin versus vancomycin or metronidazole for treating CDI in a real-world UK setting. Data were retrospectively collected from medical records of 86 patients with CDI treated with vancomycin or metronidazole at a single UK hospital between April 2011 and March 2012, and prospectively from 62 patients with CDI treated with fidaxomicin between August 2012 and July 2013. CDI cases were matched by age, financial year, and healthcare resource use to control cases. CDI recurrence rates were lower with fidaxomicin (6.5%) than vancomycin/metronidazole (19.8%). An estimated 12 additional recurrent CDIs were prevented with fidaxomicin treatment. Patients with CDI had significantly higher healthcare costs than those without CDI, with a mean excess spend of GBP 10,748 and GBP 17,451 per patient in the fidaxomicin (*p* = 0.015) and vancomycin/metronidazole cohorts (*p* < 0.001), respectively. A second CDI was associated with mean excess costs of GBP 8373 and GBP 20,249 per patient in the fidaxomicin and vancomycin/metronidazole cohorts, respectively. Despite higher fidaxomicin drug costs, overall cost savings were estimated at GBP 140,292 (GBP 2125 per CDI). In this real-world study, first-line CDI treatment with fidaxomicin reduced healthcare costs versus vancomycin/metronidazole, consistent with previous studies.

## 1. Introduction

*Clostridioides difficile* may be present in the gut of up to 18% of healthy adults [1]. Colonisation of the intestine by this pathogen is usually suppressed by the presence of native intestinal microflora; however, antibiotic-related disruption of the gut microflora increases the risk of *C. difficile* overgrowth, production of toxin and symptomatic infection [2]. Indeed, *C. difficile* infection (CDI) is the most common cause of infectious nosocomial diarrhoea, with other symptoms including severe abdominal pain, fever and leucocytosis [3]. More severe, life-threatening complications of CDI include pseudomembranous colitis, toxic megacolon and sepsis [3].

The standard treatments for CDI have been vancomycin and metronidazole; however, approximately 20–30% of patients will have a CDI recurrence within 60 days of successful treatment [4]. Fidaxomicin is a macrocyclic antibacterial agent that selectively kills *C. difficile* and significantly reduces CDI recurrence compared with vancomycin [5,6]. Furthermore, a network meta-analysis of clinical trials assessing CDI treatments concluded that fidaxomicin most frequently provides a sustained symptomatic cure and is a better first-line option than vancomycin, except for severe CDI [7]. Additionally, this analysis did not support using metronidazole as a first-line therapy for CDI [7].

The improved efficacy of fidaxomicin compared with vancomycin has been shown for sustained cure and reduction in recurrence of CDI [6,7]. As such, there has been a concentration on the costs of a recurrence of CDI. However, the higher acquisition cost of fidaxomicin means there is a need to estimate overall cost-efficacy. Of note, most CDI cost analysis studies use models that apply generalised assumptions around treatment effectiveness, healthcare resource use and costs using inputs from other published sources [8,9,10,11]. 

In the UK, CDI recurrence is associated with longer hospital stays and an estimated total cost per patient of GBP 7539–GBP 31,121 compared with GBP 6294–GBP 12,710 for primary CDI [12,13]. Among real-world studies conducted in the context of the National Health Service (NHS) in England, varying methods have been used to estimate the total costs of CDI, including a standard daily bed charge [13] and assumptions about the additional cost of a CDI based on patient admission duration after a laboratory diagnosis [12,13].

There have been relatively few studies on the cost-effectiveness of fidaxomicin [8,9,10,11]. Further real-world studies on budget justification for first line use of fidaxomicin are useful to clarify cost-effectiveness, particularly in the UK where recently updated (2021) National Institute for Health and Care Excellence (NICE) guidelines continue to recommend fidaxomicin as a second-line CDI treatment [14]. This need is made more pressing by increased pressures on healthcare resources in the wake of the coronavirus disease 2019 (COVID-19) pandemic. Indeed, COVID-19 infection is associated with several key risk factors for CDI, such as older age, broad-spectrum antibiotic use and hospitalisation [15].

In August 2012, fidaxomicin was introduced to the formulary at St George’s Healthcare NHS Trust, London, UK, for first-line treatment of CDI. A real-world budget-impact analysis of fidaxomicin was conducted 1 year after its introduction, as part of a local service evaluation required by our local Drug and Therapeutics Committee, to confirm the validity of the cost-efficacy case outlined for its use. The Patient Level Information Costing System (PLICS)–in routine use at the time–was used in this analysis. PLICS provides a more detailed assessment of inpatient costs, and thus more precise estimates of resource use, than methods used in other UK-based studies. Here, we report the results of this analysis.

## 2. Results

### 2.1. Study Flow and Patient Characteristics

From August 2012 to July 2013, 86 patients were prospectively identified by the Medical Microbiology Services laboratory as having a CDI (91 episodes in total), of whom 62 patients (62 episodes) met the inclusion criteria (fidaxomicin cohort). Five additional recurrent episodes of CDI were identified from the retrospective medical note review, of which one was excluded. Of the 62 patients with a CDI (66 episodes in total), 34 had sufficient data for the cost analysis and were able to be paired with matched controls (Figure 1). 

From April 2011 to March 2012, 118 patients were retrospectively identified by the Medical Microbiology Services laboratory as having a CDI (133 episodes in total), of whom 86 patients (97 episodes) met the inclusion criteria (vancomycin/metronidazole cohort). Another 13 CDI episodes were identified from the retrospective medical note review. Of the 86 patients with a CDI (110 episodes in total), 77 had sufficient data for the cost analysis and were able to be paired with matched controls (Figure 1).

The results of this service review were to be reported to the therapeutics committee after 1 year. As such, complete data were available only for patients discharged before the end of the financial year, meaning financial data were not available for patients included towards the end of the study period. Thus, a lower number of matched controls were identified for the fidaxomicin cohort compared with the vancomycin/metronidazole cohort (see below).

Patients in the fidaxomicin and vancomycin/metronidazole cohorts were broadly similar in terms of age, sex and concomitant antibiotic use, and in terms of the proportions with renal impairment or requiring specialist care (Table 1). The median length of hospital stay was 37 days (range: 2–188) in the fidaxomicin cohort and 26 days (range: 1–390) in the vancomycin/metronidazole cohort. Intensive care unit (ICU) care was required by 37.1% (23/62) of patients in the fidaxomicin cohort (mean stay length: 10.3 days) and 22.1% (19/86) of patients in the vancomycin/metronidazole cohort (mean stay length: 12.7 days). Death occurred within 12 days of CDI treatment in 5% (3/62) of patients in the fidaxomicin cohort and 14% (12/86) of patients in the vancomycin/metronidazole cohort.

### 2.2. CDI Recurrence

In the fidaxomicin cohort, 6.5% of patients had two CDI episodes and none had further CDI episodes. In comparison, 12.8% of patients in the vancomycin/metronidazole cohort had two CDI episodes and 7.0% had further episodes (in total, 19.8% had more than one CDI episode) (Figure 2). Recurrent CDI episodes occurred within 28 days of the end of therapy in 4.8% (3/62) of patients treated with fidaxomicin compared with 15.1% (13/86) of those who were treated with vancomycin or metronidazole (*p* = 0.039). Based on the observed number of CDI recurrences in the two cohorts, an estimated 12 CDI episodes were prevented at St George’s Hospital with fidaxomicin treatment between August 2012 and July 2013 (Table 2).

### 2.3. CDI Healthcare Costs

The median number of matched controls available per index CDI case was 4 (range: 2–14) for the fidaxomicin cohort and 16 (range: 3–73) for the vancomycin/metronidazole cohort. In the fidaxomicin cohort, costs in patients who had a CDI episode were significantly higher than in matched controls without a CDI episode, equating to a mean excess spend of GBP 10,748 per patient (*p* = 0.015) (Table 3). The mean excess cost in patients with a CDI episode (vs. matched controls without) was higher in the vancomycin/metronidazole cohort at GBP 17,451 per patient (*p* < 0.001). In the fidaxomicin and vancomycin/metronidazole cohorts, mean additional costs (in excess of those for matched controls) of a second CDI episode, compared with a first episode, were GBP 8373 and GBP 20,249 per patient, respectively (Table 4).

From April 2011 to March 2012, drug costs for the treatment of 135 CDI episodes with vancomycin or metronidazole at St George’s Hospital were GBP 4058 in total or a mean of GBP 30 per CDI episode. From August 2012 to July 2013, the drug cost for treatment with fidaxomicin at St George’s Hospital was GBP 1586 per CDI episode. Thus, treatment with fidaxomicin cost GBP 1556 more per CDI episode than vancomycin or metronidazole, or GBP 102,696 more for the entire fidaxomicin cohort. However, an estimated GBP 242,988 of in-hospital costs were saved with fidaxomicin treatment versus vancomycin or metronidazole, based on the number of CDI recurrences (*n* = 12) prevented, and the excess cost (GBP 20,249) per CDI recurrence. This equates to an overall saving of GBP 140,292 (GBP 2125 per CDI episode) once the additional drug costs for fidaxomicin treatment (GBP 102,696) are accounted for (Table 5).

## 3. Discussion

In this real-world assessment of the budget impact of CDI treatment with fidaxomicin versus vancomycin or metronidazole conducted at St George’s Hospital, CDI recurrence was reduced with fidaxomicin treatment, in keeping with previously published estimates [6,16].

This led to a substantial net reduction in overall healthcare costs of GBP 140,292 (GBP 2125 per CDI episode for the 66 episodes treated with fidaxomicin over the 12-month period). Thus, in addition to improving patient outcomes, fidaxomicin also appears to be a cost-effective first-line option for the management of CDI. These findings are consistent with those from cost-effectiveness modelling conducted in various healthcare settings, including in France, Germany, Japan and Scotland, which also indicate that first-line treatment with fidaxomicin is a cost-effective option [8,9,10,11]. Fidaxomicin was associated with an average cost saving of GBP 518 versus vancomycin for patients with a first CDI recurrence in a cost-effectiveness analysis of CDI treatment in the NHS in Scotland [8]. In another cost-effectiveness analysis of fidaxomicin versus vancomycin for first-line treatment of CDI in patients at high risk of recurrence in Germany, fidaxomicin was associated with costs per recurrence avoided of EUR 1247–EUR 2600 and reductions in the cost of treating recurrence of EUR 457–EUR 1501 per patient [11], and in a study in France, incremental costs per CDI episode avoided were EUR 2107 for first recurrence [10]. 

In the current study, primary and recurrent CDI were associated with significantly higher healthcare costs than matched controls without CDI. The mean excess total cost of CDI in patients with a single episode was GBP 9004 per patient in the fidaxomicin cohort, and GBP 13,146 per patient in the vancomycin/metronidazole cohort. The mean excess total cost of CDI in patients with two episodes was GBP 17,377 per patient in the fidaxomicin cohort and GBP 33,395 per patient in the vancomycin/metronidazole cohort. These findings are consistent with previous studies from the UK that report estimated total costs per patient of GBP 6294–GBP 12,710 for patients with one CDI and GBP 13,833–GBP 31,121 for patients with two CDIs [12,13].

Published evidence supports the use of fidaxomicin as first-line treatment for CDI, which is reflected in US and European guidelines [17,18]. The 2021 UK NICE guidelines recommend fidaxomicin as a second-line CDI treatment based on a pharmacoeconomic model illustrating a 2% probability of fidaxomicin being cost-effective compared with vancomycin, and incurring an unreasonable opportunity cost [14,19]. This analysis only considered mortality on the first decision tree of the model and used the lowest reported median cost (GBP 7539) for a CDI case [13]. In contrast, cost-effectiveness analyses that compared fidaxomicin with vancomycin or metronidazole for the treatment of CDI in several countries have consistently demonstrated lower overall healthcare costs with fidaxomicin once CDI recurrence rates are accounted for [8,9,10,11]. In particular, recurrent CDI is associated with higher healthcare resource utilization and costs than primary CDI [12,13,20,21,22,23,24,25,26]. Within the UK, CDI recurrence is associated with longer hospital stays and an estimated total cost per patient of GBP 7539–GBP 31,121 compared with GBP 6294–GBP 12,710 for primary CDI [12,13]. Of note, most CDI cost analysis studies use models that apply generalized assumptions around treatment effectiveness, healthcare resource use and costs using inputs from other published sources [8,9,10,11]. Among real-world studies conducted in the context of the National Health Service (NHS) in England, varying methods have been used to estimate the costs of CDI, including a standard daily bed charge [13] and assumptions based on patient admission duration after a laboratory diagnosis of CDI [12,13].

From 2006 to 2013, CDI infections in England declined by approximately 80% [27]. This has been largely attributed to the implementation of national control policies restricting the use of certain antibiotics, particularly fluoroquinolone [27]. Over a similar period, CDI-related mortality also declined in England, from 83.9 per million at its peak in 2007, to 14.7 per million in 2012 [28]. However, in 2020 and 2021, there has been increased interest in the potential impact of severe acute respiratory syndrome coronavirus-2 (SARS-CoV-2) infection on CDI, given that key risk factors for CDI (older age, broad-spectrum antibiotic use and hospitalization) are associated with COVID-19 [15]. Although CDI cases reported in England declined from April 2020 to March 2021 compared with previous years, this was likely due to reduced hospital activity during the first and second waves of the pandemic [29]. In addition, overlapping symptoms such as diarrhoea may, to some extent, mask true CDI case numbers [15]. Thus, the true impacts of COVID-19 on rates of CDI infection are yet to be determined.

In a phase 3 trial, 15.4% of patients treated with fidaxomicin as first-line therapy experienced a CDI recurrence compared with 25.3% of those treated with vancomycin (*p* = 0.005) [6]. This improved ‘cure’ rate with fidaxomicin is likely attributed to its ability to rapidly and selectively kill *C. difficile*, while also preserving normal anaerobic microflora that may help to protect against reinfection [30]. In contrast, other antibiotics (such as vancomycin) may merely inhibit *C. difficile* growth, while suppressing normal anaerobic microflora [30,31]. In the current real-world study, 6.5% of patients treated with fidaxomicin as first-line therapy had a CDI recurrence, compared with 19.8% of those treated with vancomycin or metronidazole. Thus, although the rate of CDI recurrence with vancomycin or metronidazole in the current real-world study was similar to that observed in the clinical trial, the magnitude of the reduction in CDI recurrence with fidaxomicin compared with vancomycin or metronidazole was substantially greater. This finding may reflect study design differences, such as the smaller numbers in our study, and the greater potential to miss recurrences through retrospective assessment of medical notes compared with prospective monitoring in a clinical trial, although the latter would be expected to affect CDI detection rates equally in both cohorts. Alternatively, it may be due to differences in the dominant *C. difficile* strains in the UK compared with the USA. For example, the implementation of national control policies in 2007 in the UK that restricted the use of broad-spectrum antibiotics may have selected strains that are less resistant to initial treatment [27].

The current study is limited by its observational nature. However, several aspects of the study design are likely to minimize such effects. First, the drugs under investigation were the only treatment options for CDI when they were investigated. As such, treatment selection was not influenced by patient characteristics. Second, matched controls were used to account for potential differences between cohorts in terms of background levels of healthcare resource use, although fewer controls were available for the fidaxomicin cohort for reasons already described. Importantly, matching was based on Healthcare Resource Group (HRG) codes, which are designed to identify groupings of patients judged to consume a similar level of resources. An additional strength of the current study was the accuracy of the cost estimates, which were based on the PLICS system and were thus specific to each ward and accurate within 3 months of each CDI admission. 

In conclusion, the lower rate of CDI recurrence associated with fidaxomicin compared with vancomycin or metronidazole as first-line treatment of CDI translates into real-world healthcare resource use savings. Given renewed pressures on healthcare resource use in the UK, optimal treatment approaches should be reflected by current treatment recommendations for CDI.

## 4. Materials and Methods

Fidaxomicin was introduced to the formulary at St George’s Healthcare NHS Trust as first-line treatment for CDI in August 2012. All patients with laboratory-confirmed CDI for their first case and first recurrence were treated with fidaxomicin. Members of the clinical microbiology team contacted the attending clinicians to advise on management and ensure adherence to the protocol.

### 4.1. Laboratory Diagnosis

Throughout the period of this service evaluation, laboratory testing for *C. difficile* did not change. It was the policy that all unformed faecal samples (>type 5 on Bristol Stool Chart [32]) from inpatients over 2 years old were tested for *C. difficile* by two-stage testing. Samples were initially tested using a glutamine dehydrogenase enzyme immune assay (EIA) (Techlab, Blacksburg, VA, USA) and positive results then confirmed by a *C. difficile* toxin EIA (Techlab ToxAB2). Only patients with faecal samples that were positive for *C. difficile* were treated with fidaxomicin.

### 4.2. CDI Cohorts

The service evaluation covered the 1-year periods before and after the introduction of fidaxomicin to compare inpatient costs in patients who received vancomycin and metronidazole with those in patients who received fidaxomicin.

CDI cases presenting at St George’s Hospital from April 2011 to March 2012 (pre-fidaxomicin) were identified through a retrospective review of Medical Microbiology Service laboratory records and medical notes (vancomycin/metronidazole cohort). CDI cases presenting at St George’s Hospital from August 2012 to July 2013 were prospectively identified by the Medical Microbiology Services laboratory (fidaxomicin cohort). 

Patients’ records (electronic and paper-based) were examined by the study team and data were extracted into an Excel spreadsheet. Electronic financial data were linked to the clinical records.

Protocols prevented testing for CDI within 28 days after an initial positive test, meaning that individuals with relapsing CDI during this period would be treated on a symptomatic basis by attending physicians, but not captured in laboratory records. To account for this, medical notes of CDI cases identified by the Medical Microbiology Services laboratory were retrospectively reviewed to identify potential CDI recurrences. Recurrent CDI episodes in the patient notes were defined as diarrhoea with clinical suspicion of CDI recurrence in patients who had been treated for CDI.

Individuals were excluded from either cohort if they were under 18 years old, did not receive treatment for CDI (or received incorrect treatment), were not treated as inpatients, or had missing medical records or financial data (Figure 1).

### 4.3. Matched Controls

Patients without CDI but with similar background healthcare resource use were identified from the St George’s Finance Department database. They were matched to individual CDI cases in each cohort by age (within 10 years of the index CDI case), financial year and the first four digits of the HRG code (e.g., GA04 = complex open, hepatobiliary or pancreatic procedures). All matched patients available for that financial year were used as controls. 

### 4.4. In-Hospital Costs

In-hospital costs were determined using PLICS (CACI Ltd., London, UK). This system allows total costs per ward to be calculated every 3 months, which are then used to determine the daily cost of a patient bed per ward for each 3-month period. The cost of a patient stay was then calculated as the cost of a patient bed per day × length of stay + high-cost diagnostics + ICU costs + high-cost drugs + surgery costs.

### 4.5. Analyses

Medians with interquartile ranges (IQRs) were used to describe data on the length of hospital stay. However, means with IQRs were used to describe cost-related data in order to reflect the total costs of patients treated, which would be underestimated with median values.

CDI episodes prevented with fidaxomicin versus metronidazole or vancomycin were estimated as the proportion of patients with observed CDI recurrences in the vancomycin/metronidazole cohort × number of patients in the fidaxomicin cohort—observed CDI recurrences in the fidaxomicin cohort.

Excess costs of CDI episodes (compared with matched controls without CDI) were calculated for each index case by subtracting the mean cost of all controls matched to that index case from the index case costs. Similar calculations were applied for data on length of hospital stay. χ^2^ testing was used to compare proportions, and t-tests or Wilcoxon signed rank tests were used to compare continuous variables as appropriate. All analyses were conducted using STATA Version 12 (StataCorp LLC, College Station, TX, USA).

### 4.6. Ethical Approval

This service evaluation was mandated as a part of the approval of fidaxomicin in St. George’s Hospital. Therefore, according to the UK Policy Framework for Health and Social Care Research, an ethical review was not required for this study. No personal information was stored in the study database. Samples were collected as part of standard care, which was unchanged during the service evaluation.

## Figures and Tables

**Figure 1 antibiotics-12-00106-f001:**
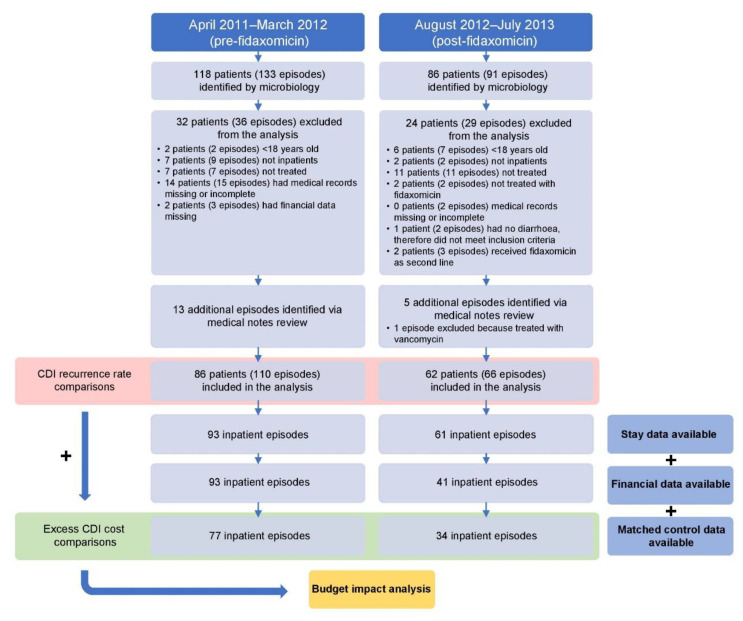
Study flow. CDI, *Clostridioides difficile* infection.

**Figure 2 antibiotics-12-00106-f002:**
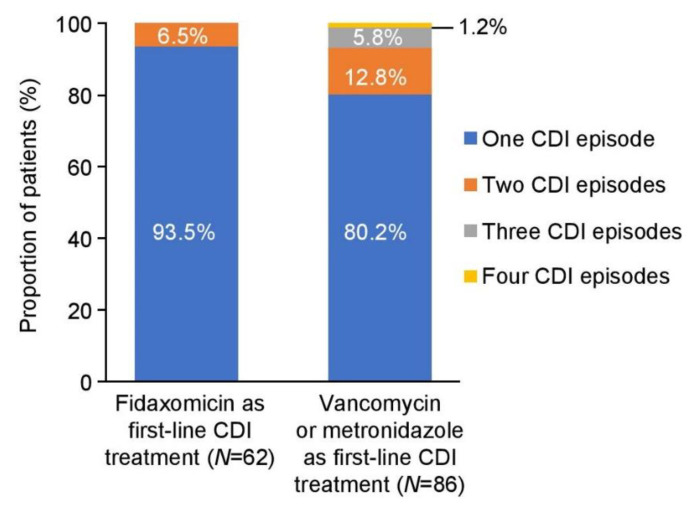
Recurrence of CDI episodes. CDI, *Clostridioides difficile* infection.

**Table 1 antibiotics-12-00106-t001:** Patient demographics and clinical characteristics.

Characteristic	Fidaxomicin (*n* = 62)	Vancomycin or Metronidazole (*n* = 86)	*p* Value
Number of CDI episodes	66	110	0.019
Median (range) age, years	77 (19–94)	78 (22–100)	0.584
Female, *n* (%)	30 (48.4)	42 (48.8)	1.000
Median (range) duration of stay, days	37 (2–188)	26 (1–390)	0.139
Renal impairment, *n* (%)	20 (32.3)	29 (34)	1.000
Severe or end-stage renal impairment, *n* (%)	11 (17.7)	18 (20.9)	0.679
Proportion with renal impairment with CDI cured, *n*/*n* (%)	19/20 (95.0)	25/29 (86.2)	0.636
Specialist care			
Haematology, *n* (%)	2 (3.2)	7 (8.1)	0.305
General surgery, *n* (%)	3 (4.8)	8 (9.3)	0.360
Oncology, *n* (%)	5 (8.1)	5 (5.8)	0.742
Geriatrics, *n* (%)	8 (12.9)	18 (20.9)	0.274
Concomitant antibiotics, *n* (%)	26 (41.9)	45 (52.3)	0.245
Proportion with concomitant antibiotics with CDI cured, *n*/*n* (%)	20/26 (76.9)	34/45 (75.6)	1.000
ICU admission, *n* (%)	23 (37.1)	19 (22.1)	0.064
Mean duration of ICU stay, days	10.3	12.7	0.715
Death within 12 days of CDI treatment, *n* (%)	3 (4.8)	12 (14.0)	0.097

CDI, *Clostridioides difficile* infection; ICU, intensive care unit.

**Table 2 antibiotics-12-00106-t002:** Estimated CDI recurrences prevented with fidaxomicin compared with vancomycin or metronidazole.

CDI Recurrence Status	Observed CDI Recurrences				Estimated CDI Recurrences Prevented with Fidaxomicin (Cumulative) ^a^
	Fidaxomicin (*n* = 62)		Vancomycin or Metronidazole (*n* = 86)		
	Patients, *n*	Rate *n*/*n*, %	Patients, *n*	Rate *n*/*n*, %	
First	4	6.5	17	19.8	8 (8)
Second	0	0.0	6	7.0	4 (12)
Third	0	0.0	1	0.2	0 (12)

CDI, *Clostridioides difficile* infection. ^a^ Calculated for each row as the proportion of patients with a recurrence in the vancomycin or metronidazole group, times the total number of patients (*n* = 62) in the fidaxomicin group, minus the actual number of recurrences that occurred in the fidaxomicin group.

**Table 3 antibiotics-12-00106-t003:** Excess costs (vs. matched controls) of CDI episodes.

Outcome	CDI Cases Treated with Fidaxomicin (*n* = 34)	CDI Cases Treated with Vancomycin or Metronidazole (*n* = 77)
Median (IQR) number of matched controls ^a^ per index CDI case	4 (2, 14)	16 (3, 73)
Median (IQR) excess duration of stay vs. matched controls, ^a^ days	13 (1, 39)	24 (6, 47)
Mean (IQR) excess cost of stay vs. matched controls, ^a^ GBP	10,748 * (−2430, 22,157)	17,451 ** (3420, 27,458)

CDI, *Clostridioides difficile* infection; HRG, Healthcare Resource Group; IQR, interquartile range. ^a^ Without CDI and matched to CDI index cases by financial year, age (within 10 years) and primary HRG code. * *p* = 0.015, ** *p* < 0.001 for excess cost vs. matched controls ^a^.

**Table 4 antibiotics-12-00106-t004:** Excess costs (vs. matched controls) of CDI episodes.

Outcome	Fidaxomicin		Vancomycin or Metronidazole	
	One CDI Episode (*n* = 28)	Two CDI Episodes (*n* = 3)	One CDI Episode (*n* = 61)	Two CDI Episodes (*n* = 10)
Median (IQR) excess duration of stay vs. matched controls,^a^ days	11 (−2, 35)	11 (9, 43)	18 (4, 38)	68 (32, 96)
Mean excess cost vs. matched controls, ^a^ GBP	9004	17,377	13,146	33,395
Mean excess cost vs. matched controls ^a^ of two (vs. one) CDI episodes, ^b^ GBP	-	8373	-	20,249

CDI, *Clostridioides difficile* infection; HRG, Healthcare Resource Group; IQR, interquartile range. ^a^ Without CDI and matched to CDI index cases by financial year, age (within 10 years) and primary HRG code. ^b^ Calculated as cost of two CDI episodes minus costs of one CDI episode.

**Table 5 antibiotics-12-00106-t005:** Estimated cost savings with fidaxomicin vs. vancomycin or metronidazole.

Variable	Description	Source	Value
A	CDI recurrences prevented	Table 2	12
B	Mean excess cost per CDI recurrence	Table 4	GBP 20,249
C	Total excess cost of CDI recurrence	A × B	GBP 242,988
D	CDI episodes treated with fidaxomicin	Figure 1	66
E	Cost of fidaxomicin per CDI episode	SGHF ^a^	GBP 1586
F	Cost of vancomycin/metronidazole per CDI episode	SGHF ^b^	GBP 30
G	Cost difference per CDI episode	E − F	GBP 1556
H	Total cost difference	D × G	GBP 102,696
I	Cost savings with fidaxomicin	C − H	GBP 140,292

CDI, *Clostridioides difficile* infection; SGHF, St George’s Hospital formulary. ^a^ August 2012–July 2013. ^b^ April 2011–March 2012.

## Data Availability

All relevant data are contained within the article.

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
