# Peer review of "Real-World Budget Impact of Fidaxomicin versus Vancomycin or Metronidazole for In-Hospital Treatment of Clostridioides difficile Infection"

_antibiotics, 2023, doi:10.3390/antibiotics12010106_

Round 1

Reviewer 1 Report

The authors have thoroughly documented the treatment of Clostridioides difficile (CDI) using vancomycin and Fidaxomicin. Fidaxomicin appears to be a first-line option for the management of CDI compared to Vancomycin or Metronidazole. This study compared the cost-effectiveness and budget impact of Fidaxomicin versus vancomycin in UK. The use of Fidaxomicin reduced CDI recurrence and the financial burden on patients. Data is very well documented using patient demographics, clinical characteristics, CDI recurrences and estimated cost savings with the use of Fidaxomicin.

Major Comments:

Major discussion in this manuscript is about the real-world assessment of the budget impact of CDI treatment. As the authors stated that data collection was between 2011 to 2013, and publishing the data decade later. Why is this study important to know a decade later, while most of the countries have already given data about the reduced cost with the use of Fidaxomicin?

As an example, to avoid the financial burden on the country, one should report the data within a couple of years of the study. On the other hand, France, Germany, Japan and Scotland data is published 2 or 3 years later.

See the cited data

Ref-8 data collected in 2010-2011 and data published in 2014 (Scotland)

Ref-11 data collected in 2014 and data published in 2016 (Germany)

This is one another manuscript similar to the reports from all other counties, and the authors should state why this data is important now?

Author Response

The author raises a fair point. We ourselves did not think such a publication would still be needed given unambiguous data supporting lower CDI recurrence rates with fidaxomicin compared with Vancomycin/Metronidazole and lower overall costs associated with its first line use to treat CDI. We note that the latest UK National Institute for Health and Care Excellence (NICE) guidelines in 2021 recommend fidaxomicin as second-line CDI treatment, and cite as their main support a costing a study undertaken in 2013 (Wilcox et al. 2017 − reference 13 in our article). The persistence of this recommendation in the UK indicates that the publication of UK-specific data such as ours is still needed. Our study is close in time to Wilcox’s 2013 data and is a more detailed in analysis. As well as adding to the evidence pool (and using more precise cost data that previous UK studies on this topic), we also hope our article will serve to reintroduce readers to already published evidence of the cost-effectiveness of using fidaxomicin as first-line treatment for CDI. We would like to point out that we do already mention these points in our discussion (lines 197−216). However, we neglected to include the information on recent NICE recommendations in the introduction. We have now included this information (lines 64−72), which we believe provides useful context for the contemporary relevance of our study. We thank the reviewer for providing the impetus to make this change and hope they consider this a satisfactory response to their critique.

Reviewer 2 Report

The article is interesting for the field and gives relevant information about fidaxomicin use. I suggest including p-values in the table where you include the sociodemographic characteristics of patients.

Author Response

We thank the reviewer for their positive assessment of our paper. As per the reviewer’s request, we have now included p values in Table 1.

Round 2

Reviewer 1 Report

Thank you for changes and I have no comments